# Light phase detection with on-chip petahertz electronic networks

Yujia Yang [1], Marco Turchetti[1], Praful Vasireddy[1], William P. Putnam [1,2,3], Oliver Karnbach[1], Alberto Nardi [1], Franz X. Kärtner[3,4], Karl K. Berggren[1] & Phillip D. Keathley [1✉]

Ultrafast, high-intensity light-matter interactions lead to optical-field-driven photocurrents with an attosecond-level temporal response. These photocurrents can be used to detect the carrier-envelope-phase (CEP) of short optical pulses, and enable optical-frequency, petahertz (PHz) electronics for high-speed information processing. Despite recent reports on optical-field-driven photocurrents in various nanoscale solid-state materials, little has been done in examining the large-scale electronic integration of these devices to improve their functionality and compactness. In this work, we demonstrate enhanced, on-chip CEP detection via optical-field-driven photocurrents in a monolithic array of electrically-connected plasmonic bow-tie nanoantennas that are contained within an area of hundreds of square microns. The technique is scalable and could potentially be used for shot-to-shot CEP tagging applications requiring orders-of-magnitude less pulse energy compared to alternative ionization-based techniques. Our results open avenues for compact time-domain, on-chip CEP detection, and inform the development of integrated circuits for PHz electronics as well as integrated platforms for attosecond and strong-field science.

[1] Research Laboratory of Electronics, Massachusetts Institute of Technology, Cambridge, MA, USA. [2] Department of Electrical and Computer Engineering, University of California, Davis, Davis, CA, USA. [3] Department of Physics and Center for Ultrafast Imaging, University of Hamburg, Hamburg, Germany. [4] Center for Free-Electron Laser Science and Deutsches Elektronen-Synchrotron (DESY), Hamburg, Germany. ✉email: pdkeat2@mit.edu

In recent years, the combination of nano-optical structures with intense, few-cycle laser sources has led to a new class of solid-state petahertz electronic devices with promising applications in time-domain metrology as well as information processing[1–19]. These petahertz devices rely on the attosecond-level temporal response of optical-field-driven photocurrents that result from the interaction of strong electric fields (tens of GV/m) with nanostructured materials[2–6,8–10,13,15–17,19–22] (for review, see refs. [23,24]). Unlike photocurrents in typical optoelectronic components, these optical-field-driven photocurrents are sensitive to changes in the electric field waveform of the optical pulse rather than the cycle-averaged photon density. Recent reports have demonstrated time-domain CEP detection with solid-state devices[6,13,15,16], and specifically, in refs. [13,15] it was shown that photoelectron emission from plasmonic nanoantennas can be used to detect changes in the carrier-envelope phase (CEP).

Consider an optical pulse with a time-dependent electric field $F(t) = F_0(t)\cos(\omega t + \varphi_{ce})$ with a carrier angular frequency $\omega$, an intensity envelope $F_0(t)$, and a CEP $\varphi_{ce}$. The CEP determines the exact optical-field waveform of the pulse and has vital importance in ultrafast and strong-field nonlinear optical processes for few- to single-cycle pulses. In the time domain, the CEP is crucial for attosecond physics including ionization of atoms and molecules[25,26], high-harmonic generation[27], and attosecond pulse generation[28,29]. For frequency-comb sources, the carrier-envelope-offset (CEO) frequency, the frequency at which the CEP is oscillating, corresponds to a shift of the comb spectrum, and is important in applications such as optical frequency synthesis[30,31], high-precision metrology[32], and quantum information science[33].

Traditionally, the detection of the CEP has been achieved with both frequency-domain interferometric techniques[31,34,35] and time-domain photoelectron emission[36–41]. These traditional CEP measurement techniques either require frequency conversion and interferometry within complex integrated or free-space systems containing multiple optical elements, or bulky vacuum apparatus and μJ-level pulses for the ionization of gas-phase atoms or molecules. A direct, time-domain CEP detection method using optical-field emission from nanoantennas[13,15] could enable shot-to-shot CEP tagging using orders of magnitude less pulse energy. Such a method holds promise for compact and on-chip CEP detectors operating in ambient conditions. However, due to: (1) low CEP-sensitivities; (2) material-damage thresholds; and (3) noise limitations; scaling the CEP-sensitive photocurrents to usable levels will require the synchronous operation of large-scale arrays of electrically-connected nanoantennas whereby the photocurrent from individual nanoantennas adds up in phase at the read-out.

In this work, we fabricated and tested large-scale networks of electrically-connected bow-tie nanoantenna pairs with nanoscale gaps for enhanced CEP detection (approximately 200 μm$^2$ array areas containing roughly 300 to 600 bow-ties). We demonstrate an order of magnitude improvement of photocurrent per emitter compared with single-triangle arrays with μm-scale emitter-collector spacing[15] when driven by the same optical source. Furthermore, we show synchronous operation of the devices across the entire array, providing a route to shot-to-shot CEP tagging of nJ-level pulses. We address key challenges in both the design and the operation of such large-scale, electrically-connected arrays including electromagnetic sensitivity to design parameters, in-situ removal of electrical shorts caused by process variations, and noise sources that limit the devices' ultimate signal-to-noise ratio. Our investigation demonstrates that we were operating the devices at or near their ultimate noise floor set by the shot-noise arising from the total number of emitted electrons. Beyond CEP detection, this work has ramifications for the development and

understanding of electrically-integrated nanoantenna devices for on-chip attosecond science and PHz electronics[7].

## Results

**Device layout and electromagnetic analysis**. Figure 1a shows a schematic of the nanoantenna device used in this work. An array of plasmonic bow-tie nanoantennas was supported by a transparent, insulating substrate. Each bow-tie nanoantenna consisted of a pair of nanotriangles. All of the left nanotriangles of the bow-ties in a column were electrically connected to one contact pad, while all right nanotriangles of that column were electrically connected to another contact pad. The device was essentially a parallelized array of photoelectron tunneling devices[13]. For a bow-tie nanoantenna, the two nanotriangles were the cathode and anode for photoelectron emission and collection respectively; and in our devices, the cathode-to-anode gap was in the range of 10–50 nm. These devices operated in ambient conditions thanks to the nanoscale cathode-to-anode gap.

We can express the net photocurrent of our configuration illustrated in Fig. 1a as $I = I_L - I_R$, where $I_L$ is the photocurrent from the left-side nanotriangles going to the right-side nanotriangles, and $I_R$ is the photocurrent from the right-side nanotriangles going to the left-side nanotriangles. Using the notation in ref. [19], we express $I_L \approx I_{0,L} + |I_{1,L}|\cos(\varphi_{ce} + \angle I_{1,L})$ and $I_R \approx I_{0,R} - |I_{1,R}|\cos(\varphi_{ce} + \angle I_{1,R})$, where $I_0$ represents the total average photocurrent, $I_1$ the complex amplitude of the first harmonic of the CEP-sensitive photocurrent, with the subscripts L and R again used to represent current coming from either the left-side or right-side nanotriangles. The negative sign in the CEP-sensitive component of $I_R$ is due to the reversal of the field polarity for the nanotriangles on the right-hand side, meaning the CEP-sensitive current is at its peak when the carrier wave is shifted by π-radians relative to the left-side nanotriangles. For perfectly symmetric nanoantenna structures, the amplitudes of the left and right contributions would be equal, meaning $I = 2|I_{1,L}|\cos(\varphi_{ce} + \angle I_{1,L})$. However, due to imperfections in fabrication, the symmetry between the current emission from both sides is not perfect, so the more general expression for the net photocurrent is $I = I_{0,\,detected} + |I_{cep}|\cos(\varphi_{ce} + \angle I_{cep})$, where $I_{0,detected}$ is the residual average total detected photocurrent, and $I_{cep}$ the complex amplitude of the first harmonic of the total detected CEP-sensitive photocurrent.

There are several advantages with the interconnected bow-tie configuration we used. First, the nanometer-scale gap ensured sub- to few-femtosecond transit times of electrons between emitters. This rapid transit time enabled hundreds of THz- to PHz-level operating bandwidths, reduced the electron's interaction with gas molecules in the ambient environment, and removed the need for large bias voltages to collect the emitted electrons. Second, by directing the photocurrent with integrated connecting wires, the signal could either be accumulated or selectively coupled to down-stream electronics on a femtosecond timescale as shown in ref. [17]. Third, the inversion symmetry of the bow-tie devices resulted in a balanced detection scheme whereby the average total detected current $I_{0,detected}$ was either significantly reduced or totally eliminated (we show later that $V_{bias}$ can be used to eliminate $I_{0,detected}$ even when the devices are not perfectly symmetric). Such a balanced scheme eliminates noise contributions that are common to both $I_{0,L}$ and $I_{0,R}$, such as noise due to laser intensity fluctuations. Furthermore, by reducing or eliminating $I_{0,detected}$, one can attach high-gain current detectors directly to the device network without risking saturation. Finally, the connecting wires and nanoantennas could all be produced with a single lithography step, simplifying fabrication

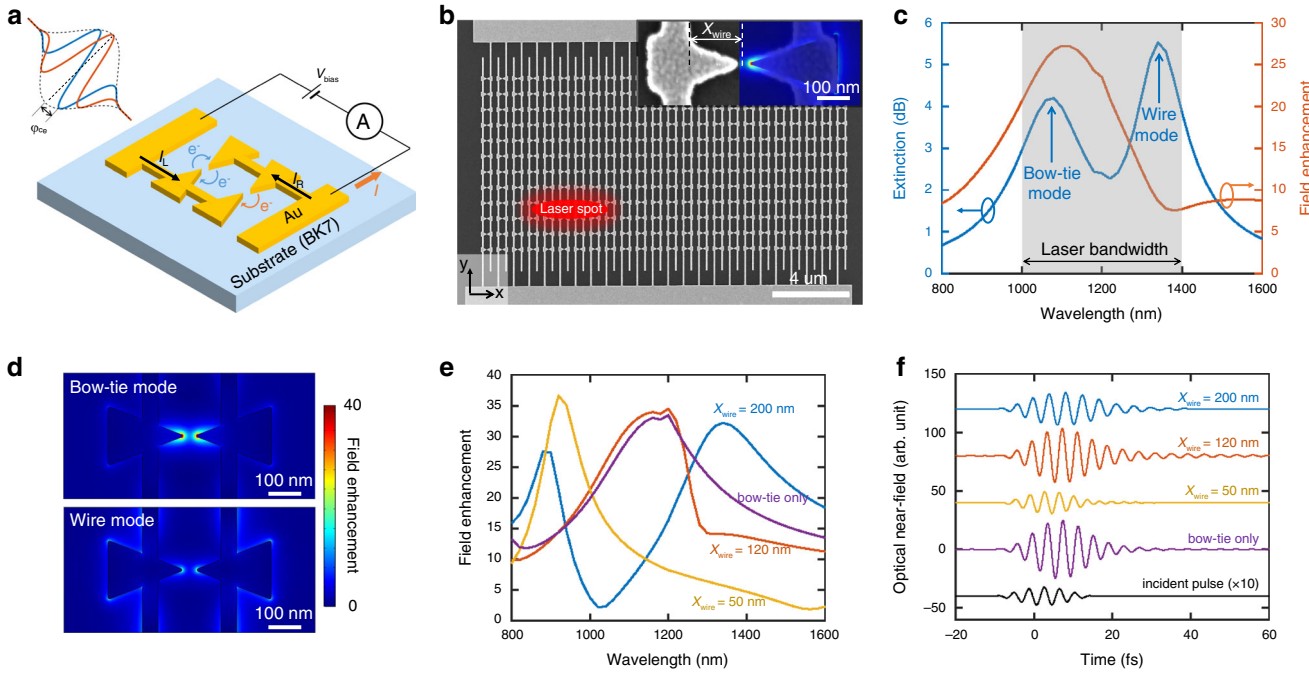

**Fig. 1 Electrically connected plasmonic bow-tie nanoantenna arrays. a** Schematic diagram of the gold bow-tie nanoantenna devices on an insulating substrate ($V_{bias}$, bias voltage; A, ampere meter; $I$, net photocurrent). Left nanotriangles are electrically connected to one contact pad, while right nanotriangles are electrically connected to the other. An incident ultrafast optical pulse induces photoelectron emission across the nano-gaps between the nanotriangles. The carrier-envelope-phase (CEP) $\varphi_{ce}$ of the pulse affects the photocurrent measured in the external circuit. For $\varphi_{ce} = \pi/2$ (blue trace), the pulse has two symmetric optical half cycles and the photocurrent in the two opposite directions cancel each other, leading to a zero net photocurrent. For $\varphi_{ce} = \pi$ (orange trace), the pulse has only one strong optical half cycle, and a net photocurrent can be measured. In the experiment, $\varphi_{ce}$ is modulated at a carrier-envelope-offset (CEO) frequency $f_{ceo}$. This CEO frequency can be measured from the photocurrent spectrum. **b** SEM image of a plasmonic nanoantenna array consisting of 288 bow-tie nanoantennas. The laser beam spot size in the experiment is also schematically illustrated. Inset: SEM image of a plasmonic bow-tie nanoantenna with a nano-gap of 28 nm. The superimposed color plot shows the simulated optical near-field profile (showing a spatial map of the electric field magnitude normalized by the incident electric field magnitude) of a nanoantenna with similar dimensions. **c** Simulated extinction spectrum (blue) and field-enhancement spectrum (orange) of the electrically connected bow-tie nanoantenna array. The two extinction peaks are labeled as the bow-tie mode and the wire mode. The gray shaded region illustrates the approximate laser bandwidth used in our experiments (see below for the measured laser spectrum). **d** Simulated optical near-field profiles (showing spatial maps of the electric field magnitude normalized by the incident electric field magnitude) of the bow-tie mode and the wire mode. The color scale is saturated for better visualization. **e** Simulated field-enhancement spectra of the plasmonic bow-tie nanoantenna arrays with different connecting wire positions ($X_{wire}$ labeled in (**b**) representing the x-distance between the inner edge of the wire and the center of the bow-tie structure). For comparison, the field-enhancement spectrum for a bow-tie nanoantenna array without the connecting wires is also shown. **f** Simulated time-domain response of the plasmonic bow-tie nanoantenna arrays with different connecting wire positions. The waveforms show the optical field at the nanotriangle tip.

and ensuring nanometer-level alignment accuracy between the emitters and connecting wires, which we will show to be critical to device operation.

The nanoantennas enhance the electric field of ultrafast optical pulses, inducing photoelectron emission. With light polarization along the bow-tie axis of symmetry in the x-direction, the optical field induces photoemission current flowing between the two nanotriangles within a bow-tie nanoantenna. The emitted electrons transit from the nanotriangles on one side to the nanotriangles on the other side, and vice versa as the optical field switches its direction. The photocurrent is then measured by connecting the two contact pads to an external ammeter and voltage bias to determine the CEP of the optical waveform. In general, the CEP is determined up to a constant offset value. This offset phase depends on several factors including the exact electron tunneling dynamics (e.g., the finite tunneling time), dispersion of the optical pulses, and the plasmonic response of the nanoantennas. As this offset phase is not essential in our experiments, we ignore it in the following discussions. We define the CEP $\varphi_{ce} = 0$ for a cosine-like optical pulse at the nanotriangle tip. The CEP $\varphi_{ce}$ of the optical waveform controls the amplitude and direction of the induced photocurrent. For example, given a

waveform represented by the blue trace in Fig. 1a, when $\varphi_{ce} = \pi/2$, the pulse has two symmetric optical half cycles, and the photocurrent in the two opposite directions cancel each other, leading to zero net photocurrent. When $\varphi_{ce} = \pi$ (represented by the orange trace in Fig. 1a), the pulse has only one strong optical half cycle (the contribution from the weak optical half cycles is negligible due to the nonlinearity of the photoemission process), and a net photocurrent is generated which flows from the left side of the excited bow-ties to the right side. Likewise when $\varphi_{ce} = 2\pi$, the same photocurrent is generated only now flowing from the right side to the left.

Figure 1b shows the SEM image of a nanoantenna array consisting of $24 \times 12$ bow-ties, with the full array covering an area of about $20 \times 10 \ \mu m^2$. The nanoantenna array was fabricated on a silicon substrate (for imaging purposes), which is conductive and free from charging issues, under the same conditions for nanoantenna fabrication with glass substrates. The inset of Fig. 1b shows a plasmonic bow-tie nanoantenna with a nano-gap of 28 nm (see Supplementary Note 1 for nano-gap size measurement). The superimposed color plot shows the simulated optical near-field profile of a nanoantenna with similar dimensions, featuring an enhanced optical field at the nano-gap.

Figure 1c, d shows the simulated optical response of a nanoantenna array. Figure 1c shows the extinction and field-enhancement spectra. The dimensions of the nanoantenna array were taken from the SEM images of a fabricated sample (see Supplementary Note 2). The extinction is defined as $-10\log_{10}(T/T_0)$, where $T$ is the power transmissivity when the nanoantenna is present, and $T_0$ is the power transmissivity when the nanoantenna is absent (but the substrate is still present). The field-enhancement is defined as the ratio of the optical near-field at the nanotriangle tip near the nano-gap (referred to as the "tip" in the following), averaged over the curved surface defined by the tip radius of curvature and gold thickness, to the optical field of the incident light. It can be seen that the extinction spectrum shows a double-peak feature with one peak at a wavelength of ~1100 nm and another peak at a wavelength of ~1300 nm. The optical near-field around the nanoantenna is plotted at 1100 nm (Fig. 1d up) and 1300 nm (Fig. 1d bottom) incident wavelengths. For the peak around 1100 nm, the optical field is localized at the bow-tie tip, indicating this peak shows the plasmonic resonant mode of the bow-tie nanoantenna, and we refer to this mode as the "bow-tie mode". For the peak around 1300 nm, the optical field is enhanced both around the bow-tie nanoantenna as well as the connecting wires, indicating a plasmon mode propagating and resonating along the periodic array of bow-ties and wires; we refer to this mode as the "wire mode". We emphasize that the wire mode is not localized and travels along the wires throughout the periodic lattice of devices, and as such depends on the periodic nature of the device layout. The wire mode has a relatively weak field enhancement at the nanotriangle tip. On the other hand, the bow-tie mode is localized to the bow-tie antenna and contributes strongly to the field-enhancement at the nanotriangle tips (see the field-enhancement spectrum in Fig. 1c and the optical near-field profiles in Fig. 1d). As a result, the bow-tie mode has a much stronger influence on the photoelectron emission compared with the wire mode.

To investigate the effect of the electrical connecting wires on the nanoantenna optical response, we performed simulations with different connecting wire positions (Fig. 1e, f). Figure 1e shows the simulated field-enhancement spectra of the plasmonic bow-tie nanoantenna arrays with different connecting wire positions ($X_{\text{wire}}$ labeled in Fig. 1b). For comparison, we also include the spectra for the bow-tie nanoantenna without connecting wires. The bow-tie nanoantenna without wires shows single-peak field-enhancement and extinction spectra (the extinction spectra are shown in Supplementary Fig. 2). When the connecting wires are added, the field-enhancement spectra splits into two peaks, corresponding to the aforementioned bow-tie and wire modes. The spectral separation of the two peaks is small and they merge into a single peak, if the connecting wire position is near the center of the nanotriangle (e.g., $X_{\text{wire}} = 120$ nm). The spectral separation of the two peaks increases, with the bow-tie mode being blue-shifted and the wire mode being red-shifted, when the connecting wire position is close to the nanotriangle base (e.g., $X_{\text{wire}} = 200$ nm) or nanotriangle tip (e.g., $X_{\text{wire}} = 50$ nm; the second peak is beyond the displayed spectral range). For the extinction spectra (Supplementary Fig. 2), a similar behavior is observed. In general, the bow-tie plasmonic mode is least disturbed when the connecting wire is close to the nanotriangle center where there is a node of the optical near-field distribution[42,43]. Figure 1f shows the simulated time-domain response at the tips of the plasmonic bow-tie nanoantennas with different connecting wire positions assuming a $\cos^2$-shaped incident pulse with a central wavelength of 1177 nm and a pulse duration (FWHM) of 10 fs (analogous to the experimental pulses we use in this work). While the broadband plasmonic enhancement preserves the ultrafast character of the incident pulse, the wire position clearly influences the time-domain profile of the waveform, and thus the photoemission response. Furthermore, the wire position affects the absolute CEP of the plasmonically enhanced optical waveforms (see Supplementary Note 2). Hence, it is critical that the wire position is both properly aligned and uniform throughout the array to ensure device response uniformity. As with the frequency-domain response, the highest field-enhancement was obtained by placing the connecting wires nears the center of the nanotriangles.

**Selective electrical breakdown of defective devices.** The fabricated bow-ties had a distribution of gap sizes that differ from the nominal size due to process variations. For nominal gap sizes on the order of 10 nm or less, it was not uncommon for several nanoantennas in a column to be connected together (i.e., no gap), with the cathode electrically shorted to the anode. This shorting made it impossible to measure any generated photocurrent. To resolve this issue we used electromigration to break the connecting wires of these shorted columns, thus, removing them from the circuit (see Fig. 2 and Supplementary Note 3). Our

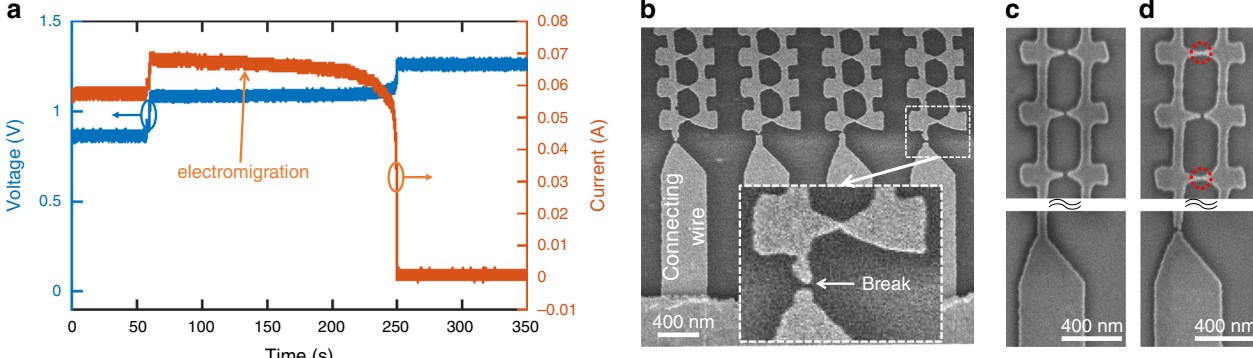

**Fig. 2 Electromigration of electrically connected nanoantenna arrays. a** Applied voltage (blue) and current (orange) across a plasmonic nanoantenna array during the electromigration process. Electromigration transformed a short-circuit array into an open-circuit array. **b** SEM image of a connected plasmonic nanoantenna array after electromigration. The wire position $X_{\text{wire}} \approx 114$ nm. The electrical connecting wires were broken and disconnected during electromigration. Inset: zoomed-in image of the connected bow-tie nanoantenna and the broken connecting wire. **c** If all the bow-tie nanoantennas along a connecting wire were disconnected to begin with, the wire was not affected and remained intact. **d** If there were shorted bow-tie nanoantennas (red dashed circles) along a connecting wire, the wire was broken via electromigration, eliminating the corresponding nanoantenna column from the functional device. In **c**&**d**, only part of the bow-tie nanoantennas along a connecting wire are shown in the image (but all the nanoantennas were inspected with the SEM).

electromigration method is reminiscent of the electrical break-down method used to remove metallic nanotubes in carbon nanotube circuits[44], and could be used in other applications using electrically connected optical nanoantennas[42,43,45].

In the electromigration process, a bias voltage was applied to the nanoantenna array creating a large current density in the wires of the shorted columns. By applying sufficient voltage to the array, and thus generating sufficiently high current densities, the connecting wires of these shorted columns were broken. However, columns having no shorted devices exhibited a very high effective resistance and were left unchanged. Figure 2a shows the voltage and current across a nanoantenna array during electromigration. The average current to initiate electromigration was ~1 mA per connecting wire, with a current density (~1 A/μm²) consistent with previous reports on current-induced electromigration in gold nanowires[46,47]. Figure 2b shows the SEM image of a shorted nanoantenna array after electromigration where the connecting wires have been broken. Figure 2c, d show non-shorted and shorted columns of bow-tie nanoantennas respectively from another array. If all bow-ties along the column were disconnected (Fig. 2c), the column was an open-circuit and the connecting wires were not broken. However, if there were connected bow-ties along the column (Fig. 2d), the column was shorted and the connecting wires were broken by electromigration. Hence, we were able to use this electromigration process as a surgical tool to selectively remove shorted devices from an array when needed.

**Measurement of ultrafast photocurrent and CEP sensitivity.** To test the CEP-sensitivity of the devices, we illuminated the nanoantenna arrays with ultrafast laser pulses from a CEP-stabilized supercontinuum source similar to that described in prior works[15,48]. The supercontinuum source has a central wavelength of ~1177 nm, pulse duration of ~10 fs full width at half maximum (FWHM) (~2.5 cycles FWHM), repetition rate of 78 MHz, and peak pulse energy of ~190 pJ. The CEO frequency $f_{ceo}$ was stabilized to 100 Hz, meaning that $\varphi_{ce}$ of each pulse was shifted by a constant amount such that $\varphi_{ce}$ of the $n$-th pulse was $\varphi_{ce}[n] = 2\pi n f_{ceo}/f_{rep} + \varphi_0$, where $n$ is the pulse number and $\varphi_0$ is a constant phase offset. The beam was focused to a spot size of 2.25 μm × 4.1 μm FWHM resulting in a peak intensity of about $2.6 \times 10^{11}$ W/cm² (corresponding to a peak field of 1.4 GV/m) before enhancement. To characterize the CEP-sensitive current

response of the device array, the current was first amplified by a transimpedance amplifier, and then the amplitude and phase of the current response at 100 Hz was measured via lock-in detection (see Supplementary Note 4). The DC bias voltage $V_{bias}$ was kept at zero unless otherwise specified. During the measurement, a barium fluoride (BaF₂) wedge was translated through the beam every 20 s providing discrete shifts (measured as 54.4° ± 11°, calculated as 57.9°) in the CEP allowing us to verify that the measured current response was indeed CEP-sensitive. The optical power absorption in the wedge is negligible. The measurement results of this scan are shown in Fig. 3.

The peak CEP-sensitive current measured was ~14 pA at 100 Hz for an array with ~1.5625 nanoantennas/μm², which corresponds to 1.12 electrons/pulse. Considering that about 11 bow-tie pairs were exposed within the FWHM of the beam spot, this corresponds to roughly 1.3 pA/bow-tie, which is similar to the results in ref. [13] (roughly 0.6 pA from a single bow-tie nanoantenna), and constitutes more than one order of magnitude increase in the total CEP-sensitive current compared with similar single-nanotriangle emitters we have reported on in prior work (up to 1.5 pA CEP-sensitive current)[15,19]. Despite these encouraging results, we observed a rather fast degradation in this current response over a period of tens of seconds before eventual stabilization to a current level of ~4 pA (~0.36 pA/bow-tie). Nevertheless, due to the combined scalability of the array configuration and benefits of the nanoscale emitter-collector separation of our devices, both the peak and the stabilized optical-field-sensitive photocurrents represent a significant improvement compared with the current generated by a single bow-tie nanoantenna[13] or plasmonic nanoparticles with mesoscopic emitter-collector gaps[15].

To demonstrate reliable operation across the entire emitter array, and the potential for signal multiplexing by interconnecting devices, we rastered the beam spot across the connected nanoantenna array while collecting the CEP-sensitive current amplitude $|I_{cep}|$ and phase $\angle I_{cep}$. These results are shown in the insets of Fig. 3. Despite active and inactive spots in the array due to the nonlinear dependence of the photoemission on the emitters' precise shape and surface properties, the scan shows that the entire array was active, with an average CEP-sensitive current response of 1.5 ± 0.8 pA. More importantly, the phase variation of the CEP-sensitive response across the entire array was only ±42° (±733 mrad) (the inset of Fig. 3a), indicating that

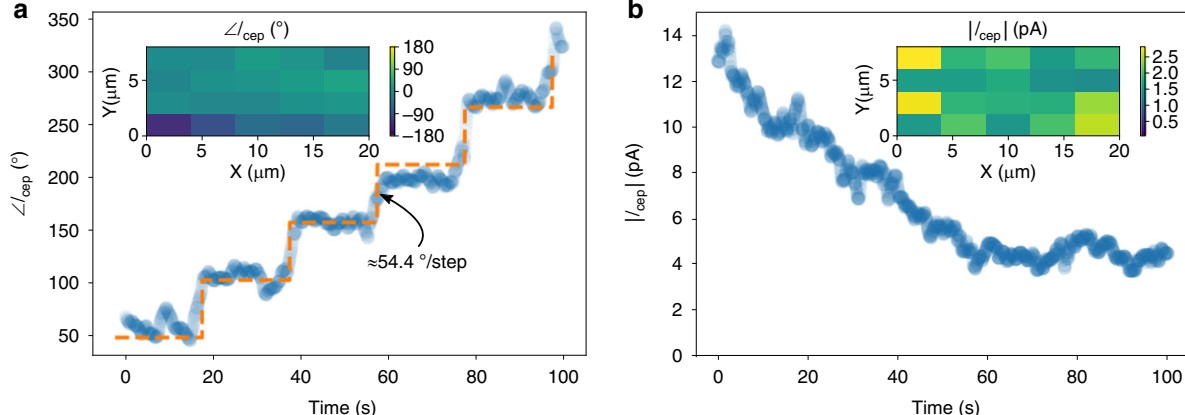

**Fig. 3 Carrier-envelope-phase-sensitive current from electrically-connected bow-tie arrays. a** Phase of $I_{cep}$ as a function of time. A barium fluoride (BaF₂) wedge is inserted into the beam every twenty seconds leading to a measured average shift in $\varphi_{ce} = 54.4° ± 11°$. The orange dashed trace shows a fit to the measured data with a staircase function. The inset shows the average phase value while scanning over the entire array area. **b** Corresponding value of $|I_{cep}|$ over the same scan shown in (**a**). The optical power absorption in the wedge is negligible. The inset shows the amplitude of $I_{cep}$ while scanning the beam over the entire array. For the nanoantenna array being tested, the wire position $X_{wire} \approx 147$ nm.

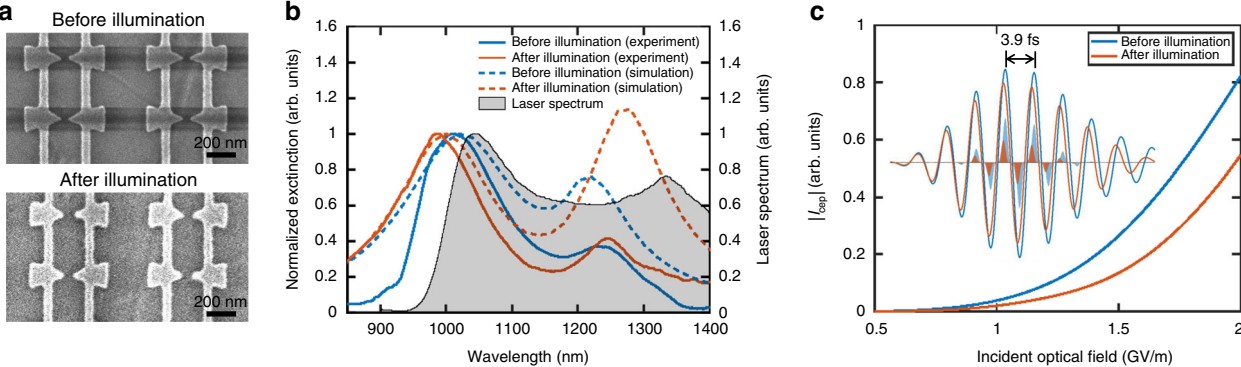

**Fig. 4 Nanoantenna device degradation during photoemission measurement. a** SEM images of a bow-tie nanoantenna array before and after photoemission measurement. The wire position $X_{wire} \approx 113$ nm. The average gap size of the bow-tie nanoantennas increased from $50.5 \pm 3.2$ nm to $62.2 \pm 11.9$ nm after illumination. The contrast variation is caused by charging issues of the insulating substrate during imaging. **b** Measured and simulated extinction spectra of the nanoantenna array shown in (**a**) before and after photoemission measurement. The gray shaded area shows the measured spectrum of the supercontinuum source. **c** Simulated CEP-sensitive photocurrent magnitude $|I_{cep}|$ vs. the optical near-field for the nanoantenna array before and after photoemission measurement. Inset: Simulated time-domain response of the nanoantenna array before and after photoemission measurement. The waveforms show the plasmonically enhanced optical fields at the nanotriangle tip for the nanoantenna arrays before and after illumination. The shaded areas show the waveforms of the photoemission current calculated from the Fowler-Nordheim theory.

by illuminating larger areas of the array while holding the peak intensity fixed (see Supplementary Note 5), one could scale the CEP-sensitive current by area with 80% efficiency (i.e., using a beam spot of $X$ greater area would result in a CEP-sensitive current increase of $0.8X$). This efficiency is calculated as $\sum_{n=1}^{N} I_{cep,n} / \sum_{n=1}^{N} |I_{cep,n}|$ for an array being scanned with $N$ laser spot locations, each of which produces a complex photocurrent $I_{cep,n}$ (the complex amplitude of the first harmonic of the CEP-sensitive current; see ref. [19]). The efficiency characterizes the degree to which the photocurrents from individual nanoantennas can add in-phase at the read-out, and is an important figure-of-merit for up-scaling the device array size. Furthermore, we point out that in the inset of Fig. 3a, the lowest row of data points differs significantly from the upper rows. This difference could be because our laser spot was close to the edge of the array. Excluding this row, we find that the phase variation reduces to $\pm 26°$ ($\pm 454$ mrad), with a scaling efficiency of 90%. Besides the CEP noise of the supercontinuum source[48] (roughly 150 mrad or 8.6°) used in the experiments, we attribute additional CEP variation to the variation of the plasmonic response of the nanoantennas, which is caused by slightly different nanoantenna geometries and wire positions due fabrication process variations (see ref. [19] and Supplementary Note 2).

We calculate that with pulse energies on the order of 10–100 nJ spread across similarly sized arrays one could achieve peak current signals of sufficient level for single-shot CEP-tagging (see Supplementary Note 6; note the calculation is based upon stabilized, rather than peak, CEP-sensitive signal to ensure stabilized operation of the device over a long time). Furthermore, we should also note that the CEP-sensitive current was measured for several samples from multiple fabrication batches, and the effects of nano-gap size and laser pulse energy were investigated (see Supplementary Note 8). The degradation of CEP-sensitive signal was observed for several devices, especially those with a small gap size illuminated with a high laser pulse energy. However, once stabilized at a lower signal level, the devices operated for hours without further degradation.

**Characterization of optically-induced device degradation.** We routinely observed the aforementioned photocurrent degradation

for small nano-gap devices under a high pulse energy (the degradation was observed for pulse energies in the range of 140–190 pJ; the exact pulse energy differs for different devices). Figure 4a shows the SEM images of a nanoantenna array before and after high-intensity illumination (roughly $10^8$ pulses with 78 MHz repetition rate and up to ~190 pJ pulse energy) for photoemission measurements. The average gap size of the bow-tie nanoantennas increased from $50.5 \pm 3.2$ nm to $62.2 \pm 11.9$ nm (see Supplementary Note 8 for measurements on more samples). To investigate the impact of this nanoantenna reshaping, we performed optical extinction measurements on nanoantenna arrays both before and after high-intensity illumination.

Figure 4b shows the simulated and measured extinction spectra from an array before and after photoemission measurements were performed. The simulation used geometries extracted from SEM images of the array (Fig. 4a). The double-peak feature of the spectra was obtained in both simulation and measurement, with the bow-tie mode around 1050 nm wavelength and the wire mode around 1250 nm wavelength. We note that the wire mode was significantly stronger in simulation than in measurement. We attribute this to the fact that the simulation used periodic boundary conditions and assumed the wires were infinitely long and the arrays perfectly periodic, while the fabricated devices consisted of finite arrays and wires with imperfect periodicity. The bow-tie mode on the other hand is associated with a localized surface plasmon resonance of the individual nanoantennas, and was thus better represented in simulation. Several other features of the extinction spectra were reproduced in simulation and measurement. For instance, the exposure to optical pulses led to a larger spectral separation between the two peaks: the bow-tie mode was slightly blueshifted, while the wire mode was slightly redshifted, with an increase in its intensity. The change of the extinction spectra before and after photoemission measurement was caused by a combined effect of laser-induced reshaping of the bow-ties and the change of the relative position between the wires and the nanotriangles (see earlier discussions) as a result of the reshaping.

The measured laser spectrum of the supercontinuum source is also shown in Fig. 4b. In the experiments, the nanoantennas were designed to have their plasmonic resonance around the blue edge, instead of the central part, of the laser spectrum. This design preserved the few-cycle character of the plasmonic near-field,

resulting in a higher CEP sensitivity[13,15]. For extinction measurement, the supercontinuum source was combined with a Ti:sapphire laser to achieve a wider spectral range for characterizing the plasmonic resonance (see Methods section).

To simulate the effect of the emitter reshaping on the CEP-sensitive photoemission, the photocurrent was estimated using a quasi-static Fowler-Nordheim tunneling theory[19]. Figure 4c shows the simulated CEP-sensitive photocurrent magnitude $|I_{cep}|$ with a varying peak incident optical-field strength for the nanoantenna array before (blue) and after (orange) photoemission measurements. The inset shows the plasmonically-enhanced optical-field waveforms (solid curves) and the calculated time-dependent photoemission current density (shaded area). This calculation confirms that a decreased field-enhancement, caused by laser-induced reshaping of the plasmonic nanostructures, is the dominant factor causing the reduced CEP-sensitive photocurrent. The result is in qualitative agreement with experimental observations. However, it is not a general result that the CEP-sensitivity always reduces as a result of the emitter reshaping as there is a complicated interplay between the CEP-sensitivity and the emitter resonance and peak intensity (see Supplementary Note 9 and ref. [19]).

**Noise characteristics of the devices.** For reliable operation of such nanoantennas as CEP detectors or optical-field-driven circuit elements, it is critical to understand their noise characteristics and how the signal-to-noise ratio (SNR) could be further enhanced for subsequent amplification and processing. As touched on earlier, a balanced detection scheme should suppress the noise associated with optical power fluctuations. However, we found that our balanced device network generated similar noise levels to comparable devices operated in an unbalanced configuration (see Supplementary Note 11), ruling out common-mode contributions as the dominant noise source even for individual emitters. Based on further analysis, we attribute the dominant part of the observed noise-floor to shot-noise from the emitted photocurrent. We found the shot-noise depended on the total emitted charge from both nanotriangle emitters in a bow-tie, rather than just the total current being detected; namely, the emitted but undetected photocurrent canceled by the balanced scheme still contributed to the device noise. As the SNR increases proportionally to the photocurrent in the shot-noise-limit, we point out further scaling of the device array as a route to improving the SNR.

To investigate the noise properties, we measured the photocurrent frequency spectrum, and investigated the noise while adjusting the operating point (balanced and unbalanced detection) by varying the DC bias voltage $V_{bias}$. A typical photocurrent frequency spectrum near $f_{ceo} = 100$ Hz is shown in Fig. 5a, exhibiting $|I_{cep}| \approx 4$ pA (before device degradation), and an SNR of ~254 (~25 dB at 0.5 Hz resolution bandwidth). To fully rule out common-mode noise contributions as the dominant noise source, we performed the photocurrent measurement with a varying DC bias voltage between the two nanotriangle emitters in the bow-tie pair. The DC bias breaks the inversion symmetry, and thus the balanced configuration of the devices. Specifically, we measured the photocurrent response at 0 Hz corresponding to the average total current detected $I_{0,detected}$ as a function of DC bias voltage $V_{bias}$. In unbalanced configurations, $I_{0,detected}$ contains a significant portion of photocurrent that depends only on the optical pulse intensity and is not sensitive to the CEP[19]. As shown in Fig. 5b, the bias voltage controlled the amount of average DC current detected from the devices by breaking the symmetry between the bow-tie pairs. Importantly, there was a bias voltage

where $I_{0,detected}$ was nearly eliminated. As one might expect, the $V_{bias}$ value that resulted in $I_{0,detected} \approx 0$ A varied slightly across the sample (sometimes slightly positive, sometimes slightly negative; see Supplementary Note 10) due to natural asymmetries from the fabrication process. Nonetheless, it was found that the noise level was insensitive to this $V_{bias}$ setting, and that the SNR of the devices tested here was strikingly similar to prior measurements performed with asymmetric single nanotriangle nanoemitters despite a reduction of $I_{0,detected}$ by more than two orders of magnitude for the case of the symmetric bow-tie pairs (see Supplementary Note 11). This similarity of the SNR indicates that the noise floor measured in both symmetric and asymmetric cases was not a result of common-mode noise in the $I_{0,detected}$ signal, e.g., noise from fluctuations of the incident optical pulse energy (see further discussion and comparison to asymmetric triangular devices in Supplementary Note 11).

To investigate the behavior of the noise-floor under illumination, we characterized the root-mean-square (RMS) average noise current $I_{noise}$ as a function of both the pulse energy (and thus the peak intensity) and $I_{0,detected}$ while setting $V_{bias} = 3$ V. (Note that we chose to operate the devices under a bias voltage such that $I_{0,detected} \neq 0$ so that it could be monitored.) For these measurements $f_{ceo}$ was unlocked, and $I_{0,detected}$ was measured by chopping the beam and measuring the current at the chopping frequency, which was set between 100–150 Hz. When examining $I_{noise}$ as a function of $I_{0,detected}$ at various frequency locations using multiple device arrays, a square-root dependence was consistently observed. Given the square-root dependence, in Fig. 5c we plot $I_{noise}$ vs. $I_{eq} = \alpha I_{0,detected}$ for two separate device arrays (referred to as Array 1 and Array 2). We define $I_{eq}$ as the equivalent shot-noise current source such that $I_{noise} = \sqrt{2q\Delta f I_{eq}}$, with $q$ being the electron charge and $\Delta f$ being the resolution bandwidth of the measurement. For Array 2 we analyze $I_{noise}$ at 100 Hz and 340 Hz for comparison. To determine $\alpha$, we fit the measured data in each case to the reference line set by $\sqrt{2q\Delta f I_{eq}}$, which is shown in orange.

The noise amplitude measured was too strong to be accounted for by the weak values of $I_{cep}$ observed, and was found to be uncorrelated to the strength of the CEP note (see Supplementary Note 11); hence, we investigated other sources of noise. Thermal noise was ruled out as the scaling of the noise with incident pulse energy was relatively uncorrelated across the tested arrays, and did not scale at the expected rate relative to the incident pulse energy (see Supplementary Note 11 for further data and explanation). In Fig. 5d we examine only the noise power spectral density as a function of frequency (corresponding to Array 2 from Fig. 5c). We note that until around 200 Hz the noise scales as $1/f^{3/2}$. This noise scaling has been observed in field-emission devices and is often attributed to work-function fluctuations resulting from Brownian motion of impurities on the metal surface[49–52]. After 200 Hz there is a transition to a spectrally flat noise response that is still well above the noise floor of the detector (Supplementary Fig. 12). We note that this transition to the spectrally flat region can shift from sample to sample and spot to spot (see for instance Supplementary Fig. 12 where this transition is closer to 150 Hz) but appears to be a general behavior of the noise spectrum from our devices. This spectrally flat noise response means that simply shifting to higher values of $f_{ceo}$ would not significantly improve the SNR of the devices, and that, due to the observed square-root dependence of the noise on the detected current, such electrically connected networks and larger beam spots for increased signal are critical to improving the SNR.

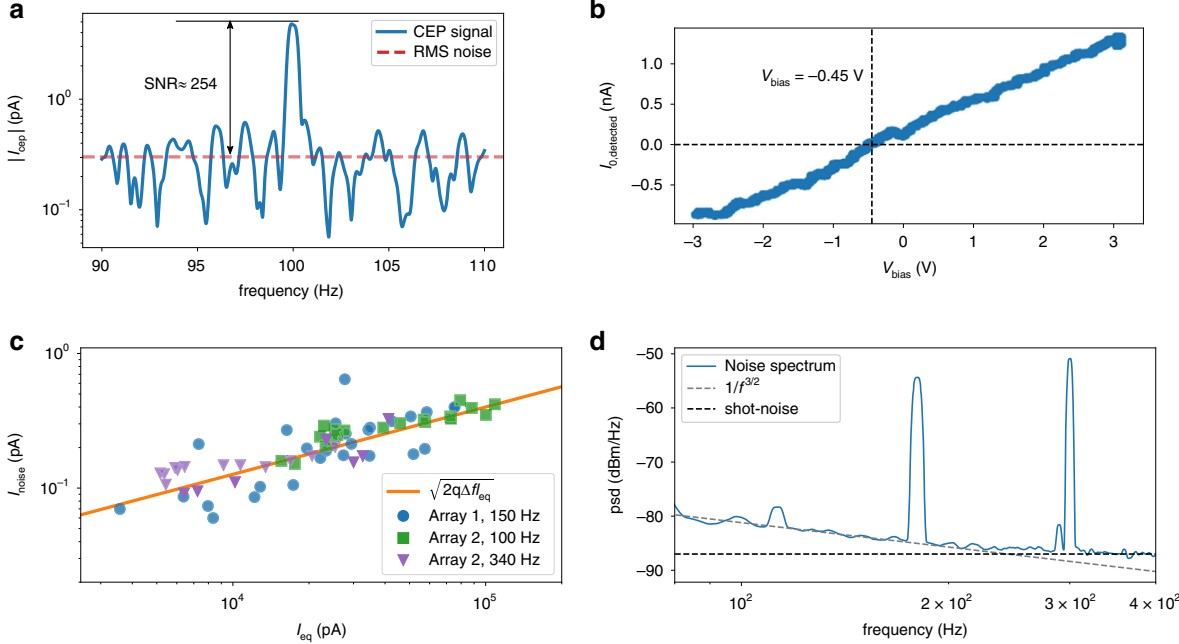

**Fig. 5 Results demonstrating SNR, balanced configuration, and noise characterization. a** Spectrum of the photocurrent near $f_{ceo} = 100$ Hz, showing the CEP-sensitive current spike and surrounding noise with an SNR of ~254 at 0.5 Hz resolution bandwidth. The nanoantenna array being tested has a wire position $X_{wire} \approx 155$ nm. **b** Plot of $I_{0,detected}$ demonstrating the use of $V_{bias}$ to null the average total detected current signal. The signal is nulled when $V_{bias} \approx -0.45$ V. **c** Plot of the noise signal $I_{noise}$ vs. the equivalent shot noise current $I_{eq}$ for two arrays while setting $V_{bias} = 3$ V. Blue dots represent results from Array 1 at 150 Hz, the green dots represent Array 2 at 100 Hz, and the purple dots represent Array 2 at 340 Hz. The orange reference curve represents $\sqrt{2q\Delta f I_{eq}}$. **d** Power spectral density (psd) as a function of frequency for Array 2 of **c**. The CEP was unlocked. Reference curves demarcate $1/f^{3/2}$ scaling and the shot-noise floor respectively. The visible notes are due to background power-line noise (at 120 Hz, 180 Hz, and 300 Hz, respectively).

We attribute the flat spectral region in Fig. 5d to the shot-noise floor of the devices arising from the total emitted current $I_{0,emitted}$ (that is the sum of all charge emitted from every nanotriangle emitter on the sample surface, including the charge that is not detected due to the cancellation of the net photocurrent by balanced detection). Since the shot-noise of electron emission from independent nanotriangle emitters in bow-tie pairs would not be correlated, and would thus not cancel despite the inversion-symmetric bow-tie arrangement, the RMS average shot noise current should scale with the total emitted current such that $I_{eq} = I_{0,emitted}$ in this region, i.e., $I_{noise} = \sqrt{2qI_{0,emitted}\Delta f}$ (of course, as we discuss in Supplementary Note 7, this may not hold for cases where the emission from both sides would be correlated, such as from a single quantum electron state extending across the gap). The value of $I_{0,emitted}$ and the precise CEP sensitivity can vary from device to device, and might explain the shift in the transition frequency from $1/f^{3/2}$ scaling to shot-noise from device to device.

To support our interpretation that shot-noise from $I_{0,emitted}$ was the cause of the observed flat noise-floor beyond 200 Hz, we used the data from 340 Hz on Array 2 shown in Fig. 5c to estimate that $I_{0,emitted} \approx 40$ nA at the highest pulse energy tested. Given the strength of the CEP-sensitive photocurrent for this array at the peak tested pulse energy, we calculate a CEP-sensitivity $I_{cep}/I_{0,emitted}$ between $10^{-4}$ and $10^{-5}$. Both the estimated $I_{0,emitted}$ and CEP-sensitivity values are in good agreement with prior results using single nanotriangle emitters made of the same material with similar tip geometries, peak enhancement factors and the same laser source, lending extra confidence to the conclusion that the SNR of the CEP response of devices was measured either just above or at the shot-noise-limit[15,19]. This shot-noise-limit emphasizes the importance of using large scale device arrays, both for improving the signal, and

the SNR, which should increase with $\sqrt{I_{0,emitted}}$. Considering the illumination of an entire array having dimensions of $50 \times 50\ \mu m^2$, we find that it would be possible to achieve an SNR of 20–30 dB at a resolution bandwidth of 1 kHz (see Supplementary Note 6). With further improvement of the detector area or the nanoantenna density, the improved SNR could be sufficient for feedback and control of $f_{ceo}$.

## Discussion

In this work, we have demonstrated on-chip medium-to-large-scale integration of nanoscale optical-field-driven devices. We have investigated device design, fabrication, multiplexing, degradation, and noise characteristics. In the current design, nearly identical individual devices were electrically connected in parallel and operated synchronously to produce a large optical-field-sensitive photocurrent. Noise analysis shows that we were operating these devices near their shot-noise limit which arises due to the average total emitted current signal from each emitter in the bow-tie pairs despite the fact that there is no net detected average total signal. Our findings emphasize the need for large-scale arrays such as those investigated here to further improve the overall SNR and achieve sufficient photocurrents for shot-to-shot CEP tagging. To that end, we find that by illuminating arrays of similar dimensions to those currently fabricated with pulses having an energy of just 10–100 nJ and similar duration to that used in our experiments, one could achieve an improvement in SNR of two to three orders of magnitude and sufficient photocurrent for shot-to-shot CEP tagging. Compared with current state-of-the-art field-ionization CEP detectors[37], such devices reduce the needed pulse energy by at least two to three orders of magnitude (see Supplementary Note 6), while replacing bulky vacuum equipment and electron detectors with a simple,

monolithic optoelectronic device operating in ambient conditions. These performance improvements could have a significant impact on experiments and applications requiring optical waveform control and synthesis. In order to operate the CEP detectors at even higher power for potentially greater signal yield and SNR without device degradation, alternative refractory plasmonic materials[53,54] could be considered. In addition, the nanoantenna arrays demonstrated in this work can be viewed as optical-field-driven, PHz integrated circuits with identical individual devices. By modifying the interconnection and integrating heterogeneous devices and materials, this study will inform the development of more complex integrated circuits of PHz electronics[24,55,56], as well as on-chip integrated platforms for attosecond and strong-field science.

## Methods

**Device fabrication**. The plasmonic bow-tie nanoantenna arrays were fabricated on glass (BK7) substrates (*MTI Corp.*) with electron beam lithography (EBL) and a metal liftoff process. A ~70 nm film of poly(methyl methacrylate) (PMMA) (*MicroChem Corp.*) was spin-coated onto the substrate and then soft-baked at 180 °C. A thin layer of Espacer (*Showa Denko*) was then spin-coated for charge dissipation during EBL. Bow-tie nanoantenna and electrical connecting wire patterns were produced by an Elionix F-125 EBL system using an accelerating voltage of 125 kV and a beam current of 500 pA. The bow-tie nanostructures and electrical connecting wires were defined and fabricated in one EBL step instead of two aligned EBL steps. Fabrication of the two structures together ensured good alignment accuracy between the bow-ties and the connecting wires, which is critical for tuning the optical response of the nanoantenna arrays as shown in the main text. After exposure, Espacer was removed with 60 s DI water rinse. Exposed PMMA was developed in 3:1 isopropyl alcohol (IPA): methyl isobutyl ketone at 0 °C for 30 s and then dried with flowing nitrogen gas. The 2 nm Ti and 20 nm Au were then deposited via electron-beam evaporation. Metal lift-off was performed in n-methylpyrrolidone (NMP) at 60 °C for ~60 min during which the sample was gently rinsed with flowing NMP. The lift-off was finished with 15 min sonication. No damage to the nanostructures was observed after sonication. The union of multiple connected bow-ties formed a larger structure compared with the isolated nanoparticles, making the bow-ties unaffected by sonication. After lift-off, the sample was rinsed with acetone and IPA. Finally, gentle oxygen plasma ashing (50 W, 60 s) was applied to remove residual resist and solvents. The contact pads were fabricated via a subsequent photolithography step. Positive-tone photoresist S1813 (*Shipley*) was spin-coated and soft-baked at 110 °C for 4 min. Photolithography was performed with a Heidelberg μPG 101 direct laser writing system. After exposure, the samples were developed in Microposit MF-321 developer for 90 s and in deionized water for 15 s. The 20 nm Ti and 80 nm Au were then deposited via electron-beam evaporation. Lift-off was performed by soaking the samples in acetone for ~30 min followed by 3 min sonication. Nanoantenna arrays consisting of 24 × 12 or 24 × 24 bow-ties were fabricated and tested. The array pitch was 800 nm in the x-direction (the direction of the bow-tie long axis), and 800 nm or 400 nm in the y-direction (the direction of the bow-tie short axis), with the full array covering an area of about $20 \times 10 \ \mu m^2$. For the nanotriangles, the nominal altitude was 260 nm, and the nominal base width was varied from 155 nm to 235 nm for tuning the nanoantenna plasmonic resonance. The bow-tie nano-gap size was tuned by changing the lithographic dose, with the smallest average gap size below 20 nm. The nominal linewidth for the connecting wires was 50 nm. The nominal thickness was 20 nm for both the nanoantennas and the connecting wires.

**Electromigration**. The electromigration process was similar to the one described in ref. [46] originally used for the fabrication of metallic electrodes with nanometer separation. In our electromigration process, a bias voltage was applied to the nanoantenna array connected in series with a 2.5 Ω resistor. A small resistor ensured a small change of the voltage across the nanoantenna device during electromigration when a constant bias voltage was used. The bias voltage and the voltage (hence the current) across the resistor were monitored by an oscilloscope. The shorted devices had a low resistance and hence a high current, which broke the connecting wires of these devices via electromigration. The normal devices had a large resistance and negligible current, and remained intact after the electromigration process. As an example, Fig. 2a shows the voltage and current across a nanoantenna array during electromigration. The applied voltage (across the array and the resistor) was kept at 1 V for 50 s, and then increased to 1.25 V. Initially, the nanoantenna array was shorted, with a resistance of 15 Ω. Electromigration process started at 50 s, showing a decrease of the current, which indicates an increasing resistance. At 250 s, the current dropped to zero, implying the array was transformed into an open-circuit.

**Electromagnetic simulation**. We simulated the optical response of the plasmonic nanoantenna arrays with a finite element method electromagnetic solver (*COMSOL Multiphysics*). The modeled nanoantenna geometry was taken from layout design parameters or SEM images of fabricated nanostructures. The 20-nm-thick gold nanoantenna was placed on the interface between vacuum and a glass (BK7) substrate, with a 2-nm-thick Ti adhesion layer in between. The optical properties of Au and Ti were taken from the work by Johnson and Christy[57] describing optical constants of the metals fabricated under similar conditions to ours (vacuum-evaporated polycrystalline thin films). The refractive index of glass was fixed at 1.5 as its dispersion was negligible in the wavelength range of interest. Periodic boundary conditions were applied to the simulation domain boundaries to model a periodic array of nanoantennas. The array pitch was 800 nm in the bow-tie long-axis direction and 400 nm in the bow-tie short-axis direction. The plane-wave light was incident normally with a linear polarization along the bow-tie long-axis to excite the plasmonic mode. Perfect matched layers were added to the top and bottom of the simulation domain to absorb outgoing electromagnetic waves and model semi-infinite vacuum and substrate. Extinction and field-enhancement were evaluated in the frequency-domain. The extinction is defined as $-10\log_{10}(T/T_0)$, where $T$ is the power transmissivity when the nanoantenna is present, and $T_0$ is the power transmissivity when the nanoantenna is absent (but the substrate is still present). The field-enhancement is defined as the ratio of the optical near-field at the nanotriangle tip near the nano-gap, averaged over the curved surface defined by the tip radius of curvature and gold thickness, to the optical field of the incident light. For the time-domain response, a $\cos^2$-shaped incident pulse with a central wavelength of 1177 nm and a pulse duration of 10 fs FWHM was assumed. The spectrum of the pulse was obtained by a Fourier transform. Broadband (800–1600 nm wavelength) frequency-domain simulations were performed to evaluate the enhanced optical near-field at the nanotriangle tip. The field-enhancement was assumed to be unity for wavelengths outside the simulation spectral range. The time-domain response was obtained by an inverse Fourier transform of the frequency-domain response. The CEP-sensitive photocurrent was estimated by a harmonic analysis of the Fowler–Nordheim photoemission current induced by the transient optical field[19].

**Experimental setup**. The nanoantenna devices were exposed to a few-cycle, CEP-stabilized optical pulse train from a supercontinuum-based fiber laser source[48]. The supercontinuum source has a central wavelength of ~1177 nm, pulse duration of ~10 fs FWHM (~2.5 cycles FWHM), repetition rate of 78 MHz, and peak pulse energy of ~190 pJ. The laser beam was focused to a spot size of 2.25 μm × 4.1 μm FWHM resulting in a peak intensity of about $2.6 \times 10^{11}$ W/cm² before plasmonic enhancement. In the experiments, the CEO frequency $f_{ceo}$ was stabilized to 100 Hz by a local oscillator. The CEP RMS noise of the laser was about 150 mrad.

In the external circuit, the photocurrent generated by the nanoantenna device was first amplified by a transimpedance amplifier (*FEMTO*) and then detected by a lock-in amplifier (*Stanford Research Systems*) using the CEO frequency as the reference frequency (see Supplementary Note 4). Discrete shifts of the CEP were introduced by the mismatch of the group and phase velocities in the $BaF_2$ wedge (2 mm thickness and 0.75° angle), which was inserted by 2.5 mm every 20 s and led to a CEP shift calculated as 57.9°. The CEP-response across entire nanoantenna arrays were measured by scanning the piezoelectric sample stage while simultaneously recording the photocurrent magnitude and phase. The photocurrent spectra were measured by a vector signal analyzer (*Agilent*). The extinction spectra of the samples were measured by a fiber-coupled optical spectrum analyzer (*Ando*). For extinction measurement, the supercontinuum laser source was combined with a Ti:sapphire laser to achieve a total spectral range of roughly 0.7–1.4 μm in wavelength. The measurements of $I_{0,detected}$ were performed by chopping the beam and using lock-in detection to measure the amplitude and phase of the first harmonic of the chopped photocurrent signal.

## Data availability

All data in this study are available from the authors upon reasonable request. Source data are provided with this paper.

## Code availability

The codes used for simulations and display items are available from the authors upon reasonable request.

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

## Acknowledgements

This material is based upon work supported by the Air Force Office of Scientific Research under award numbers FA9550-19-1-0065, and FA9550-18-1-0436. F.X.K. acknowledges support by the European Research Council under the European Union's Seventh Framework Programme (FP7/2007–2013) through the Synergy Grant 'Frontiers in Attosecond X-ray Science: Imaging and Spectroscopy' (AXSIS) (609920) and by the Cluster of Excellence 'CUI: Advanced Imaging of Matter' of the Deutsche Forschungsgemeinschaft (DFG) - EXC 2056 - project ID 390715994. This work was also supported through the PIER Hamburg – MIT Program.

## Author contributions

Y.Y., W.P.P., F.X.K., K.K.B., and P.D.K. conceived the concept and designed the experiments. Y.Y. and P.D.K. performed the electromagnetic analysis, fabricated the devices, and tested electromigration. Y.Y., M.T., P.V., O.K., and P.D.K. performed photocurrent measurements and contributed to experimental data analysis. M.T. and P.D.K. performed the extinction measurements. Y.Y. and A.N. analyzed the laser-induced reshaping of the nanoantennas. P.V. and P.D.K. analyzed the noise characteristics of the devices. All authors interpreted the results and contributed to the writing of the manuscript.

## Competing interests

The authors declare no competing interests.
