## [Peer Review File · Nature Communications]

REVIEWER COMMENTS

Reviewer #1 (Remarks to the Author):

Yang et al. present an experimental study of a nano-optical device capable of detecting the carrier-envelope phase (CEP) of ultrashort laser pulses. When an ultrashort pulse irradiates a bow-tie antenna structure, the resulting photocurrent crossing the vacuum gap between two opposing nanoantennas is strongly sensitive to the waveform of the pulse. A small CEP-sensitive current results, which can be detected using a lock-in scheme. Using on-chip arrays of bow-tie antennas, the authors demonstrate that a sizable CEP-sensitive current results, up to an order of magnitude higher than a similar demonstration from the same research group (Putnam et al., 2017).

The manuscript is well-written and technically sound. Its conclusions are based on strong experimental evidence. In particular, I commend the authors for presenting a very careful study, also discussing results, reasons and solutions when the device doesn't operate at its best. Up-scaling of nano-circuitry is challenging, and the authors describe many aspects of this challenge and how they were able overcome it, with several pathways towards improving the device performance in the future. The manuscript definitely deserves publication. It also shows novelty with respect to prior work:

- * Enhancement of the CEP-sensitive current by one order of magnitude.
- * Design and fabrication of a large array device.
- * Practical pathway to circum-navigate fabrication errors.
- * Assessment of laser-induced damage and its impact on device performance.
- * First discussion of noise characteristics in such a device, to the best of my knowledge.

In my opinion, the novelty meets the readership and quality level of Nature Communications, so I recommend the publication of this manuscript. However, two minor things should be discussed explicitly In the manuscript:

- * The device does not measure the CEP of the laser pulses directly, but only up to an (experimentally unknown) value. Indeed, there is nothing like an absolute phase of a laser pulse, but a CEP can be defined locally for a point in space. This should be clarified in the text.
- * From the simulations of the plasmonic field, which CEP shift would be expected between the plasmonic field (in the presence of the nanostructure) and the incident field (around the same point of space, but in the absence of the nanostructure)?

I also recommend to include the following (new) references related to the work:

- * Zimmermann et al., Nano Lett. 19, 1172 (2019)
- * Sederberg et al., Nature Communications 11, 430 (2020)
- * Ludwig et al., Nature Physics doi: 10.1038/s41567-019-0745-8 (2020)
- * Kubullek et al., Optica 7, 35 (2020)

Reviewer #2 (Remarks to the Author):

In this manuscript, the authors demonstrate CEP detection using optical-field-driven electron emission in an array of plasmonic nanoantennas. Compared with the authors' previous report of Ref.15, the authors integrate plasmonic nanoantennas monolithically. This integration is essential for improving the overall SNR. In fact, they operate the nanoantenna devices near their shot-noise limit. The direct detection of the light phase with on-chip device has tremendous importance for future PHz electronics as well as CEP detection. In addition, the contents are highly innovative. Therefore the reviewer thinks that this paper may be accepted for publication after some revisions. Please refer to the comments shown below.

- 1) The authors should show the power spectrum of the incident pulse in Fig.1c or Fig.1e, to compare it with the antenna resonances.
- 2) The authors show in Fig.1e that the field enhancement spectrum differs significantly depending on the connecting wire position X_{wire} and that highest field-enhancement is obtained by placing the connecting wires near the center of the nanotriangles. The authors should explicitly write X_{wire} used for the successive experiments shown in Figs 2-5.
- 3) In page 9, the second paragraph, the authors write that there is the phase variation of 42 degree of the CEP-sensitive response across the entire array. It would be much informative if the authors could explain why there exist such phase variation.
- 4) In page 9, the second paragraph, the authors also write "one could scale the CEP-sensitive current by area with 80% efficiency." The authors should explain how they deduce this conclusion logically. Why is the efficiency 80%?
- 5) Are the experimental data shown in Figs. 5a-d taken after the nanoantenna devices are degraded or before degraded?

We would like to thank the referees for their time and effort in reviewing our manuscript. We have read through their comments, and we provide our point-by-point responses to these comments in the document below.

To make it easy to navigate, we have numbered the pages here and provided a table of contents below. We also used the following formatting:

- Black text indicates a referee comment.
- Light-blue text indicates our response to a referee comment.
- Red text indicates a reference to specific changes made in the text to address the referee's comment.

Sincerely,

Y. Yang and P. D. Keathley on behalf of all authors

Table of Contents

Table of Contents	1
Referee 1	1
Referee 2	4
Summary of Changes to the Manuscript	7

Referee 1

Yang et al. present an experimental study of a nano-optical device capable of detecting the carrier-envelope phase (CEP) of ultrashort laser pulses. When an ultrashort pulse irradiates a bow-tie antenna structure, the resulting photocurrent crossing the vacuum gap between two opposing nanoantennas is strongly sensitive to the waveform of the pulse. A small CEP-sensitive current results, which can be detected using a lock-in scheme. Using on-chip arrays of bow-tie antennas, the authors demonstrate that a sizable CEP-sensitive current results, up to an order of magnitude higher than a similar demonstration from the same research group (Putnam et al., 2017).

The manuscript is well-written and technically sound. Its conclusions are based on strong experimental evidence. In particular, I commend the authors for presenting a very careful study, also discussing results, reasons and solutions when the device doesn't operate at its best. Up-scaling of nano-circuitry is challenging, and the authors describe many aspects of this challenge and how they were able overcome it, with several pathways towards improving the device performance in the future. The manuscript definitely deserves publication. It also shows novelty with respect to prior work:

- * Enhancement of the CEP-sensitive current by one order of magnitude.
- * Design and fabrication of a large array device.
- * Practical pathway to circum-navigate fabrication errors.
- * Assessment of laser-induced damage and its impact on device performance.
- * First discussion of noise characteristics in such a device, to the best of my knowledge.

In my opinion, the novelty meets the readership and quality level of Nature Communications, so I recommend the publication of this manuscript. However, two minor things should be discussed explicitly in the manuscript:

We thank the reviewer for his/her comments regarding the impact and novelty of our work.

- * The device does not measure the CEP of the laser pulses directly, but only up to an (experimentally unknown) value. Indeed, there is nothing like an absolute phase of a laser pulse, but a CEP can be defined locally for a point in space. This should be clarified in the text.

We thank the reviewer for pointing out the need for a definition of the absolute CEP. We have clarified that the CEP is determined up to a constant offset value, and added the definition for zero CEP.

In the "Device layout and electromagnetic analysis" section, when explaining the device operation mechanism, we now clarify the CEP is only determined up to a constant offset value. We also add a definition of zero CEP for a cosine-like optical pulse at the nanotriangle tip.

- * From the simulations of the plasmonic field, which CEP shift would be expected between the plasmonic field (in the presence of the nanostructure) and the incident field (around the same point of space, but in the absence of the nanostructure)?

In Fig. 1f, we show the incident optical waveform together with the simulated plasmonic near-field waveforms. The plasmonic response of the nanoantenna induces ringing of the near-field after excitation, as well as an offset of the absolute CEP. This offset value depends on the specific plasmonic resonance, hence affected by the nano-gap size, the field strength, and the wire position. In the inset of Fig. 4c, we compare the simulated plasmonic near-field waveforms for nanoantennas before and after laser reshaping. The laser reshaping leads to different nano-gap sizes and field strengths, resulting in a CEP offset.

We have added the discussions on the CEP offset in Supplementary Note 2. We have added Supplementary Fig. 3 (also shown below), which shows the simulated normalized electric field waveforms and the corresponding CEP offset values for the optical waveforms shown in Fig. 1f. As discussed in the manuscript, the wire position strongly affects the plasmonic response, and hence affects the absolute CEP value. For an incident pulse with its CEP set to 0, the plasmonically enhanced waveforms have CEP shifts depending on the nanoantenna plasmonic response. The simulated absolute CEP values are -0.36π , -0.14π , -1.33π , and -0.35π for $X_{\text{wire}} = 200$ nm, 120 nm, 50 nm, and bow-tie without the wire, respectively. Our simulated CEP shift is in agreement with the results shown in Ref.22 of the manuscript. We have also pointed out the plasmonic response as one of the causes for the absolute CEP offset when addressing the previous comment.

Supplementary Figure 3. Simulated optical waveforms at the tip of the plasmonic bow-tie nanoantenna arrays with different connecting wire positions (X_{wire} labeled in Fig. 1b). **a**, Normalized waveforms for bow-tie nanoantennas with $X_{\text{wire}} = 200$ nm, 120 nm, and 50 nm, together with the waveform for bow-tie nanoantenna without the wire and the incident waveform. **b-f**, Each of the waveforms shown in **a**. The absolute CEP of each waveform is labeled.

Experimental determination of the absolute CEP is beyond the scope of this work as we only measure the relative shift of the CEP. However, further investigation of the CEP offset and the effect of the plasmonic response could be important in future work involving single-shot or interferometric autocorrelation measurement.

In the "Device layout and electromagnetic analysis" section, we mention the absolute CEP is affected by the plasmonic response and hence the wire position, and refer to Supplementary

Note 2. In the "Supplementary Note 2", we add Supplementary Fig.3 and corresponding discussions on the simulated CEP shift induced by the plasmonic response and wire position.

I also recommend to include the following (new) references related to the work:

- * Zimmermann et al., Nano Lett. 19, 1172 (2019)
- * Sederberg et al., Nature Communications 11, 430 (2020)
- * Ludwig et al., Nature Physics doi: 10.1038/s41567-019-0745-8 (2020)
- * Kubullek et al., Optica 7, 35 (2020)

We have now included the above references.

Referee 2

In this manuscript, the authors demonstrate CEP detection using optical-field-driven electron emission in an array of plasmonic nanoantennas. Compared with the authors' previous report of Ref.15, the authors integrate plasmonic nanoantennas monolithically. This integration is essential for improving the overall SNR. In fact, they operate the nanoantenna devices near their shot-noise limit. The direct detection of the light phase with on-chip device has tremendous importance for future PHz electronics as well as CEP detection. In addition, the contents are highly innovative. Therefore the reviewer thinks that this paper may be accepted for publication after some revisions. Please refer to the comments shown below.

We would first like to thank the reviewer for his/her comments.

1) The authors should show the power spectrum of the incident pulse in Fig.1c or Fig.1e, to compare it with the antenna resonances.

We thank the reviewer for pointing out the need for a power spectrum of the laser source or the incident pulse. We have added an illustration in Fig. 1c to show the laser bandwidth used in our experiments. We have also added the measured laser spectrum of the supercontinuum source in Fig. 4b. We choose Fig. 4b to display the measured laser spectrum as the figure also shows the measured and simulated extinction spectra that characterize the nanoantenna resonance used in the experimental portions of the paper, while Fig. 1c is more for demonstration purposes. We have also explained that a relatively high CEP sensitivity could be achieved by designing the nanoantenna resonance around the blue edge of the laser spectrum, according to Refs.13,15 of the manuscript. Furthermore, to eliminate possible confusion, we have added a

notion that the extinction spectra were measured by combining the supercontinuum laser with a Ti:sapphire laser to achieve a wider spectral coverage.

In Fig. 4b, we now include the laser spectrum of the supercontinuum source. We also explain that a relatively high CEP sensitivity can be achieved by designing the nanoantenna resonance around the blue edge of the laser spectrum, according to Refs.13,15 of the manuscript. Furthermore, we clarify that the extinction spectra were measured by combining the supercontinuum laser with a Ti:sapphire laser (this is also mentioned in the Methods section).

2) The authors show in Fig.1e that the field enhancement spectrum differs significantly depending on the connecting wire position X_{wire} and that highest field-enhancement is obtained by placing the connecting wires near the center of the nanotriangles. The authors should explicitly write X_{wire} used for the successive experiments shown in Figs 2-5.

In the caption of Figs. 2-5, we now specify the X_{wire} values for the nanoantennas.

3) In page 9, the second paragraph, the authors write that there is the phase variation of 42 degree of the CEP-sensitive response across the entire array. It would be much informative if the authors could explain why there exist such phase variation.

The CEP noise of the supercontinuum source used in the experiments is about 150 mrad (9 degrees) (for more information see Ref.48 of the manuscript). We attribute additional CEP variation to the variation of the plasmonic response of the nanoantennas, which is caused by slightly different nanoantenna geometries and wire positions due fabrication process variations. In Supplementary Note 2, we have added Supplementary Fig. 3 (also shown below), which shows the simulated normalized electric field waveforms and the corresponding CEP offset values for the optical waveforms shown in Fig. 1f. As discussed in the manuscript, the wire position strongly affects the plasmonic response, and hence affects the absolute CEP value. For an incident pulse with its CEP set to 0, the plasmonically enhanced waveforms have CEP shifts depending on the nanoantenna plasmonic response. The simulated absolute CEP values are -0.36π rad (-64.8 degrees), -0.14π rad (-25.2 degrees), -1.33π rad (-239.4 degrees), and -0.35π rad (-63 degrees) for $X_{\text{wire}} = 200$ nm, 120 nm, 50 nm, and bow-tie without the wire, respectively. Therefore, for nanoantennas within the device array, slightly different wire positions could lead to a detected local CEP variation across the array. Furthermore, nanoantenna geometry variations lead to different plasmonic responses and hence optical field strengths in the nano-gaps. As demonstrated in Ref.19 of the manuscript, the peak field strength at the nanoantenna tip affects the phase of the CEP-sensitive photocurrent reading and thus would contribute to variation in the measured phase in the CEP-sensitive response as the beam is rastered across the array. Hence, the nanoantenna geometry variation also contributes to the CEP variation.

Supplementary Figure 3. Simulated optical waveforms at the tip of the plasmonic bow-tie nanoantenna arrays with different connecting wire positions (X_{wire} labeled in Fig. 1b). **a**, Normalized waveforms for bow-tie nanoantennas with $X_{\text{wire}} = 200$ nm, 120 nm, and 50 nm, together with the waveform for bow-tie nanoantenna without the wire and the incident waveform. **b-f**, Each of the waveforms shown in **a**. The absolute CEP of each waveform is labeled.

In the "Measurement of ultrafast photocurrent and CEP sensitivity" section, we explain the CEP variation is caused by the variation of plasmonic response due to fabrication process variations, and refer to Supplementary Note 2. In the "Supplementary Note 2", we add Supplementary Fig.3 and corresponding discussions on the simulated CEP shift induced by the plasmonic response and wire position.

4) In page 9, the second paragraph, the authors also write "one could scale the CEP-sensitive current by area with 80% efficiency." The authors should explain how they deduce this conclusion logically. Why is the efficiency 80%?

This scaling efficiency is calculated as $\left| \frac{\sum_{n=1}^N I_{cep,n}}{\sum_{n=1}^N |I_{cep,n}|} \right|$ for an array being scanned with N laser spot locations, each of which produces a complex photocurrent $I_{cep,n}$ (the complex amplitude of the first harmonic of the CEP-sensitive current; see Ref.19 of the manuscript). This efficiency characterizes the degree to which the photocurrents from individual nanoantennas add in-phase at the read-out, and is a figure-of-merit for up-scaling the device array size.

In the "Measurement of ultrafast photocurrent and CEP sensitivity" section, we now clarify the definition and deduction of the scaling efficiency for a device array.

5) Are the experimental data shown in Figs. 5a-d taken after the nanoantenna devices are degraded or before degraded?

The data in Fig. 5 was taken before device degradation.

In the "Noise characteristics of the device" section, when referring to Fig. 5a, we mention the data was taken before device degradation.

Summary of Changes to the Manuscript

In the following list, we summarize key changes to the manuscript. We exclude any minor corrections, such as spelling errors or figure numbering corrections. All changes in the manuscript text are highlighted in red color.

Manuscript Body

- Device layout and electromagnetic analysis
 - We clarified the CEP is determined up to a constant offset value.
 - We added the definition of zero CEP.
 - We mentioned the absolute CEP was affected by the wire position, and referred to Supplementary Note 2.
 - We added an illustration of the laser bandwidth to Fig. 1c.
- Measurement of ultrafast photocurrent and CEP sensitivity
 - We clarified the definition and deduction of the scaling efficiency for a device array.
- Characterization of optically induced device degradation
 - We added the measured laser spectrum to Fig. 4b.
- In the caption of Figs. 2-5, we now specify the X_wire values for the nanoantennas

Methods

- We clarified that the extinction measurement was performed with the supercontinuum laser and a Ti:sapphire laser to achieve a total spectral range of roughly 700-1400 nm in wavelength.
- The Methods section is now in front of the References section.

Data availability

- This section was added.

Code availability

- This section was added.

References

- New references were added: Ref.20, Ref.21, Ref.22

Supplementary Information

- In Supplementary Note 2, we added discussions on the effect of plasmonic response on the absolute CEP. Specifically, we added the simulation results showing the wire position affected the absolute CEP of the plasmonically enhanced optical waveforms.
- In Supplementary Note 6, we extended discussions on future avenues to scale the devices for single-shot CEP tagging. We noted the single-shot operation could be achieved by scaling the device array size, and improving the CEP-sensitivity by using, for instance, sub-two-cycle optical pulses similar to those used in Refs.6,13,16 of the manuscript.
- We added Supplementary Figure 6 in accordance with the revised Supplementary Note 6.

REVIEWERS' COMMENTS

Reviewer #1 (Remarks to the Author):

The authors have addressed the comments and concerns from me and the other referee in a satisfactory way. I recommend the manuscript "as is" for publication in Nature Communications.

Reviewer #2 (Remarks to the Author):

The manuscript has been revised in a careful manner according to the reviewer's comments, and it can be now accepted for publication in Nature Communications.

Reviewer #1 (Remarks to the Author):

The authors have addressed the comments and concerns from me and the other referee in a satisfactory way. I recommend the manuscript "as is" for publication in Nature Communications.

Reviewer #2 (Remarks to the Author):

The manuscript has been revised in a careful manner according to the reviewer's comments, and it can be now accepted for publication in Nature Communications.

We would like to thank the reviewers again for their time and effort in reviewing our manuscript. By addressing the reviewers' comments, we believe the review process has greatly improved the quality of the manuscript.

Sincerely,

Y. Yang and P. D. Keathley on behalf of all authors